# Effects of the Water Extract of Fermented Rice Bran on Liver Damage and Intestinal Injury in Aged Rats with High-Fat Diet Feeding

**DOI:** 10.3390/plants11050607

**Published:** 2022-02-24

**Authors:** Ting-Yu Chen, Ya-Ling Chen, Wan-Chun Chiu, Chiu-Li Yeh, Yu-Tang Tung, Hitoshi Shirakawa, Wei-Tzu Liao, Suh-Ching Yang

**Affiliations:** 1Graduate Institute of Metabolism and Obesity Sciences, Taipei Medical University, Taipei 11031, Taiwan; ma48109006@tmu.edu.tw; 2School of Nutrition and Health Sciences, Taipei Medical University, Taipei 11031, Taiwan; ylchen01@tmu.edu.tw (Y.-L.C.); wanchun@tmu.edu.tw (W.-C.C.); clyeh@tmu.edu.tw (C.-L.Y.); 3Graduate Institute of Biotechnology, National Chung Hsing University, Taichung 40227, Taiwan; peggytung@nchu.edu.tw; 4Laboratory of Nutrition, Graduate School of Agricultural Science, Tohoku University, Sendai 980-8857, Japan; shirakah@m.tohoku.ac.jp; 5Chian-E Biomedical Technology Corporation, Taipei 11031, Taiwan; cindyxox.tw@gmail.com; 6Research Center of Geriatric Nutrition, College of Nutrition, Taipei Medical University, Taipei 11031, Taiwan; 7Nutrition Research Center, Taipei Medical University Hospital, Taipei 11031, Taiwan

**Keywords:** water extract of fermented rice bran, non-alcoholic fatty liver disease, high-fat diet, aged rat

## Abstract

The purpose of this study was to investigate the protective effects of the water extract of fermented rice bran (FRB) on liver damage and intestinal injury in old rats fed a high-fat (HF) diet. Rice bran (RB) was fermented with *Aspergillus kawachii*, and FRB was produced based on a previous study. Male Sprague Dawley rats at 36 weeks of age were allowed free access to a standard rodent diet and water for 8 weeks of acclimation then randomly divided into four groups (six rats/group), including a normal control (NC) group (normal diet), HF group (HF diet; 60% of total calories from fat), HF + 1% FRB group (HF diet + 1% FRB *w/w*), and HF + 5% FRB group (HF diet + 5% FRB *w/w*). It was found that the antioxidant ability of FRB was significantly increased when compared to RB. After 8 weeks of feeding, the HF group exhibited liver damage including an increased non-alcoholic fatty liver disease score (hepatic steatosis and inflammation) and higher interleukin (IL)-1β levels, while these were attenuated in the FRB-treated groups. Elevated plasma leptin levels were also found in the HF group, but the level was down-regulated by FRB treatment. An altered gut microbiotic composition was observed in the HF group, while beneficial bacteria including of the Lactobacillaceae and Lachnospiraceae had increased after FRB supplementation. In conclusion, it was found that FRB had higher anti-oxidative ability and showed the potential for preventing liver damage induced by a HF diet, which might be achieved through regulating imbalanced adipokines and maintaining a healthier microbiotic composition.

## 1. Introduction

The World Health Organization (WHO) reported that more than 1 billion people globally were 60 years and older in 2020 [1], and over 1.9 billion adults were overweight and obese (with a body-mass index (BMI) ≥25 kg/m^2^) in 2016 [2]. According to data from the 2013–2016 Nutrition and Health Survey in Taiwan (NAHSIT), the prevalence of being overweight and obese (defined as a BMI of ≥24 kg/m^2^) and having metabolic syndrome (METS) were more elevated in the elderly than in adults [3]. There were also a number of risk factors and chronic disease signs caused by or related to obesity which were ranked in the top ten leading causes of death in Taiwan [4]. These conditions revealed that the incidence of chronic diseases such as obesity and METS had greatly increased with aging.

Non-alcoholic fatty liver disease (NAFLD) is considered the most common chronic liver disease in the world, which affects about one-quarter of the world’s population [5] and 11.4~41% of the population in Taiwan [6]. The two-hit hypothesis is commonly used to describe the pathophysiology of NAFLD. The first hit is mainly correlated with insulin resistance (IR), which stimulates the release of free fatty acids (FAs; FFAs) from adipocytes, thereby leading to excess accumulation of triglycerides (TGs) in the liver [7]. Oxidative stress triggers the second hit of NAFLD, accompanied by the generation of lipid peroxidation and proinflammatory cytokines such as tumor necrosis factor (TNF)-α, interleukin (IL)-1β, and IL-6, and endoplasmic reticulum (ER) stress, which accelerate the progression of NAFLD [8]. The mechanisms of dysregulated lipid metabolism play key roles in NAFLD progression. In NAFLD, decreased adiponectin levels and increased leptin levels result in an adipokine imbalance [9]. Adiponectin is an adipocyte-derived anti-inflammatory adipokine, which is expressed on hepatic membranes by adiponectin receptor 2 (AdipoR2) and increases activation of adenosine monophosphate-activated protein kinase α (AMPKα) and the nicotinamide adenine dinucleotide-dependent deacetylase, sirtuin-1 (SIRT1) [10]. It was reported that decreased serum adiponectin levels and liver AdipoR2 protein levels were observed after high-fat (HF) diet intake, which inactivates AMPKα and SIRT1, thereby inhibiting FA oxidation [11]. In addition, a lower expression of peroxisome proliferator activated receptor (PPAR)-α was found in NAFLD, causing the reduced expression of carnitine palmitoyltransferase (CPT)-1, which in turn inhibits mitochondrial β-oxidation [12]. Sterol regulatory element-binding protein (SREBP)-1c, mainly a lipogenic transcription factor which is usually increased in NAFLD, can up-regulate expressions of lipogenesis-associated enzymes such as stearoyl coenzyme A desaturase 1 (SCD1), FA synthase (FAS), and acetyl-CoA carboxylase (ACC) [13,14]. On the other hand, gut dysbiosis is also recognized as a crucial risk factor in NAFLD, which causes dysfunction of the gut endothelial barrier, elevation of intestinal permeability, and enabled translocation of gut-derived endotoxins, thereby mediating liver inflammation [15]. In addition, bacterial diversity, beneficial microorganisms, and the availability of total short-chain FAs (SCFAs) are reduced in the gut microbiota of the elderly [16]. Moreover, dietary pattern is an important factor in determining the composition of the intestinal microbiome [17]. A previous study showed that the intake of dietary fiber increased Prevotella abundances and the production of gut metabolites (e.g., SCFAs) [18], while a HF diet resulted in an increase in the Bacteroides enterotype [19] and the Firmicutes/Bacteroidetes (F/B) ratio [20]. Consequently, nutrients appear to be involved in modulating the gut microbiotic structure in order to balance changes in the intestinal microbiota in the elderly.

Rice bran (RB), which contributes about 10% to the whole grain weight, is a by-product of the rice milling process [21]. In Taiwan, 110,000~140,000 tons of RB are produced annually in the milling process. However, most RB is used as animal feed or is discarded as an agricultural waste. Utilizing agricultural byproducts or wastes including RB is a recent trend of reducing the environmental burden and saving resources [22]. During the past few years, previous studies reported that RB is rich in bioactive components, including vitamin E, vitamin B complex, γ-oryzanol, ferulic acid, plant sterols, phytic acid, among others, which possess antioxidant, anti-inflammatory, antihyperlipidemic, and hepatoprotective activities [23,24]. Studies also indicated that fermented RB (FRB) can effectively enhance nutritional values, such as protein, dietary fiber, and total phenolic contents compared to non-FRB [25]. Hence, RB has gradually been developed into a low-cost and functional food ingredient. The potential protective effects of FRB against metabolic disorders were investigated in several studies, including METS, hypertension [25], diabetes [26], inflammatory bowel disease (IBD) [27,28], and tumor progression [29], which validated the functional properties of FRB in improving lipid metabolism impairment, regulating glucose levels [25], suppressing oxidative stress [28], attenuating inflammatory responses [26,28], and affecting the gut microbiota [27]. These effects are also involved in regulating NAFLD development. However, the ameliorating effects of FRB on NAFLD are still controversial. Possible reasons include differences in fermentation processes, bacterial species, and animal models.

Based on the above background, this study established a HF diet-fed aged rat model to simulate the conditions of obesity and metabolic disorders in senior citizens. However, the effect of RB from Taiwan fermented by *Aspergillus kawachii* has not been discussed in aged rats with HF diet consumption. Therefore, as shown in Figure 1, the purpose of this study was to investigate whether FRB can attenuate HF diet-induced NAFLD development in aged rats, as well as explore the underlying mechanisms of the protective effects of FRB against NAFLD.

## 2. Results

### 2.1. Antioxidant Ability of FRB

As shown in Table 1, FRB had higher antioxidant capacity than RB. Additionally, the radical-scavenging activity in FRB was higher than that of RB.

### 2.2. Food Intake and Final Body Weights (BWs)

Food intake showed no differences among the groups (Figure 2A). Additionally, the HF group had significantly higher caloric intake and food efficiency ratio than the NC group, while there were no differences between the HF group and FRB treatment groups (Figure 2B,C). BW gain and final BWs of the HF group were significantly higher than those of the NC group. However, BW gain and final BWs of the FRB treatment groups only showed a lower trend than the HF group (Figure 2D,E).

### 2.3. Liver Damage Indicators

#### 2.3.1. Liver Function Index

Compared to the NC group, the HF group only showed a trend of higher plasma aspartate aminotransferase (AST) and alanine aminotransferase (ALT) activities. Meanwhile, only a decreasing trend was observed in the FRB treatment groups when compared to the HF group (Figure 3).

#### 2.3.2. Liver Histopathological Examinations

Liver steatosis and inflammatory cell infiltration in the HF group were elevated relative to the NC group, whereas the extents of steatosis and inflammation were ameliorated by the FRB intervention (Figure 4A). Moreover, the HF group showed a significantly higher NAFLD score compared to the NC group. In addition, the NAFLD score was significantly reduced with FRB supplementation, especially in the HF+5% FRB group, compared to the HF group (Figure 4B, Table 2).

#### 2.3.3. Liver Cytokine Levels

There were no significant differences among all groups in hepatic IL-6 and IL-10 levels. Compared to the NC group, the hepatic TNF-α level was significantly higher; however, no change was observed in hepatic TNF-α levels among the HF group and FRB treatment groups. Moreover, the hepatic IL-1β level of the HF group was slightly higher than that of the NC group. Conversely, hepatic IL-1β levels of the HF + 1% FRB and HF + 5% FRB groups were significantly lower than that of the HF group (Table 3).

#### 2.3.4. Plasma and Liver Lipid Peroxidation

No significant difference was found in liver thiobarbituric acid-reactive substance (TBARS) levels. Compared to the NC group, a slight increase was seen in the plasma TBARS level in the HF group, but the FRB treatment groups only showed a trend of lower plasma TBARS levels compared to the HF group (Table 4).

### 2.4. Lipid Metabolism Indicators

#### 2.4.1. Hepatic Total Cholesterol (TC) and TG Concentrations

As shown in Table 5, significant differences were not observed in hepatic TC concentrations among all groups. The HF group displayed a slightly higher liver TG level compared to the NC group, while FRB-treated groups showed a tendency of decreased liver TG levels compared to the HF group.

#### 2.4.2. Hepatic Lipid Metabolism-Related Protein Expressions

There were no significant changes in liver AMPKα, phosphorylated (p)-AMPKα/AMPKα, leptin receptor, or adipoR2 protein expressions among all groups (Figure 5B,D–F). SIRT1 and p-AMPKα protein expressions were significantly elevated in the HF group compared to the NC group, while there were no differences between the HF group and FRB treatment groups (Figure 5A,C).

#### 2.4.3. Hepatic Fatty Acid Metabolism-Related Gene Messenger (m)RNA Levels

There were no significant differences in mRNA levels of SREBP-1c, ACC, SCD1, FAS or PPARα among these groups. In addition, CPT-1 mRNA levels showed no difference between the NC and HF groups, while it was significantly lower in the HF + 1% FRB group than that of the HF group. On the other hand, mRNA levels of SREBP-1c and ACC were slightly higher in the HF group compared to the NC group, but the HF + 1% FRB and HF + 5% FRB groups had a decreasing trend of SREBP-1c and ACC levels compared to the HF group (Table 6).

#### 2.4.4. Plasma Adipokine Levels

Compared to the NC group, plasma leptin levels were significantly higher in the HF group. However, FRB-treated groups showed a decreasing trend of plasma leptin levels compared to the HF group (Figure 6A). No differences were found in plasma adiponectin levels or adiponectin/leptin ratios (Figure 6B,C).

### 2.5. Blood Glucose Regulators

According to the results of the IR analysis presented in Figure 7, there were no differences in fasting blood glucose levels, fasting insulin levels, or homeostasis model assessment of the IR index (HOMA-IRI) among all groups.

### 2.6. Intestinal Damage Indicators

#### 2.6.1. Intestinal Tight Junction Protein mRNA Levels

As shown in Figure 8, the intestinal mRNA levels of zonula occludens (ZO)-1 and occludin presented no differences among all groups. There was also no change in intestinal claudin-1 mRNA levels between the NC and HF groups; however, the intestinal claudin-1 mRNA level was significantly lower in the HF + 1% FRB group compared to the HF group.

#### 2.6.2. Fecal Microbiotic Analysis

##### The Firmicutes-to-Bacteroidetes (F/B) Ratio and Alpha-Diversity Index

There were no significant differences in the F/B ratio among all groups (Figure 9A). In order to determine alterations in the fecal microbiotic community structure, an alpha-diversity analysis was conducted. The Chao1 and ACE indices commonly describe the species richness, while Shannon and Simpson indices refer to species diversity. There were no changes in the fecal microbiotic richness among all groups (Figure 9B). In addition, no differences were found in the Shannon and Simpson indices between the NC and HF groups. However, the HF + 1% FRB group showed significantly higher levels of the Shannon and Simpson indices compared to the HF group (Figure 9C).

##### Beta-Diversity Index

To assess the variations in the fecal microbiota, a beta-diversity analysis was performed using a principal coordinate analysis (PCoA) plot. As shown in Figure 10, the PCoA showed different microbiotic distributions between the HF and FRB treatment groups.

##### Linear Discriminant Analysis of the Effect Size (LEfSe)

The LEfSe approach and linear discriminant analysis (LDA) score were used to identify changes in bacterial taxonomical abundances among groups. It was found that the RF39 (order) and *Anaeroplasma* (genus) of the Tenericutes phylum and the Ruminococcaceae (family) and *Lactococcus garvieae* (species) of the Firmicutes phylum were abundant in the NC group (Figure 11). When comparing differences between the NC and HF groups, pathogenic bacteria such as the Proteobacteria (phylum), Peptococcaceae (family), *rc4_4* (genus), *Oscillospira* (genus), and *Ruminococcus gnavus* (species), which belong to the Firmicutes phylum, were overrepresented in the HF group (Figure 12). When comparing differences between the HF group and FRB-treated groups, SCFA-producing bacteria such as *Lactobacillales* (order), *Lactobacillaceae* (family), Lachnospiraceae (family), *Blautia* (genus), and *Blautia producta* (species) of the Firmicutes phylum were overexpressed in the HF + 1% FRB group (Figure 13). Moreover, the dominant bacteria in the HF + 5% FRB group also contained the SCFA-producing bacteria, including Lactobacillales (order), *Lactobacillaceae* (family), *Lachnospiraceae* (family), *Coprococcus* (genus) and *Lactobacillus* (genus) of the Firmicutes phylum as well as *Bacteroides uniformis* (species) of the Bacteroidetes phylum (Figure 14).

#### 2.6.3. Fecal SCFA Concentrations

There were no significant differences in isobutyric acid, butyric acid, isovaleric acid, valeric acid, 4-methylvaleric acid, or hexanoic acid levels among all groups. The heptanoic acid level was significantly lower in the HF group than in the NC group, whereas no change was seen between the HF group and FRB-treated groups. Moreover, compared to the NC group, the HF group showed a trend of a lower propionic acid level; however, only an increasing trend was observed in the HF + 1% FRB and HF + 5% FRB groups compared to the HF group (Table 7).

## 3. Discussion

### 3.1. Food Intake and BWs

HF diets are widely used in animal models to induce the development of NAFLD, simulating clinical conditions of METS, hyperlipidemia, obesity, and IR [30]. It was reported that patients over 50 years of age with diabetes or obesity had a 66% risk of experiencing non-alcoholic steatohepatitis (NASH) with advanced fibrosis [31]. In the present study, as shown in Figure 2, HF-diet-fed old rats had significantly higher caloric intake and final BWs than normal-diet-fed old rats, while FRB supplementation showed a trend to decrease the final BWs.

### 3.2. Liver Damage

After feeding the HF diet for 8 weeks, hepatic steatosis and inflammatory cell infiltration as well as the higher TNF-α level were observed (Figure 4, Table 2 and Table 3). Plasma AST, ALT, and TBARS levels and the hepatic IL-1β level were only slightly increased (Figure 3, Table 3 and Table 4). However, based on the liver histopathological examinations, the NAFLD score was significantly elevated. Thus, these results showed that HF-diet-induced liver damage was induced [32] although the liver damage might be mild in this study. As reported previously, TNF-α, IL-6, and IL-1 cytokine family members are stimulated by an HF diet; however, anti-inflammatory adipokines such as adiponectin and IL-10 were shown to have decreased [14]. On the other hand, Liu et al. reported that old rats fed a HF diet for 12 weeks presented a significant increase in serum ALT levels and NAFLD activity scores [33]. Nunes-Souza et al. also found that plasma AST and ALT levels, hepatic fat deposition as well as plasma and hepatic lipid peroxidation were significantly elevated by the 14-week HF diet feeding in aged mice [34]. In this study, the HF diet feeding period was only 8 weeks. The shorter feeding period might be the reason why plasma AST and ALT levels, and plasma and hepatic lipid peroxidation did not significantly change.

When supplemented with FRB, liver damage was alleviated as described by significantly lower NAFLD scores and liver IL-1β levels, and slightly lower levels of liver TNF-α and plasma TBARSs (Figure 4, Table 2, Table 3 and Table 4). Likewise, Ai et al. found that FRB could mitigate hepatic steatosis in mice with type 2 diabetes mellitus (T2DM) [35]. Lipid accumulation was also improved by FRB treatment in IR-HepG2 cells [35]. In a mouse model of obesity, it was reported that supplementation with FRB could decrease fatty depositions in liver tissues [36]. Another study showed that the inflammatory reactions were inhibited after FRB treatment by suppressing the mRNA levels of TNF-α and IL-6 in RBL-2H3 cells [37]. Consequently, the present results supported the protective effects of FRB on HF diet-induced liver damage possibly due to ameliorating hepatic steatosis and cytokines in aged rats.

### 3.3. Lipid and Glucose Metabolism

In this study, no differences were found in hepatic TC or TG levels among the groups. The hepatic TG level only showed an increasing trend in the HF group compared to the NC group (Table 5). It was found that hepatic lipids were significantly elevated, and IR was observed in young rats fed the HF diet for 8 weeks [38]. Thus, age might be a factor that caused no change in hepatic TC and TG levels. In future studies, a young control group should be designed as a control group due to the age-related differences.

This study demonstrated that plasma leptin levels and hepatic SIRT1 and p-AMPKα protein expressions were significantly higher in rats fed the HF diet for 8 weeks, while the plasma adiponectin/leptin ratio exhibited a downward trend (Figure 5 and Figure 6). It was pointed out that hypoadiponectinemia and higher levels of serum leptin commonly occurred in NASH patients [9]. Increased serum leptin levels and decreased serum adiponectin levels were also observed in mice fed a HF diet for 10 weeks [36]. In addition, SIRT1 and AMPK were indicated to suppress lipogenesis through inhibition of SREBP-1 levels and also activation of FA oxidation via PPARα in the liver [39]. Chronic HF diet consumption reduced AMPK activity in liver tissues, which in turn led to inhibition of FA oxidation, followed by metabolic inflammation, oxidative stress, and IR [40]. However, results showed a reverse trend of hepatic SIRT1 and p-AMPKα protein expressions compared to previous studies. It was suspected that the aged rats may increase the activities of catabolic enzymes after HF diet intake in a compensatory manner, but the underlying mechanisms should be explored in future studies. Furthermore, mRNA levels of hepatic FA metabolism-related genes did not show a difference after 8 weeks of HF dietary intake in the present study (Table 6). Although hepatic lipogenic genes such as *SREBP-1c* and *ACC* only had upward trends in the HF group, hepatic steatosis was found in the HF group according to hepatic histopathology. Expressions of these hepatic FA metabolism-related proteins must be measured in the future study.

Previous studies suggested that FRB exerted anti-obesity and anti-dyslipidemic effects on HF diet-fed mice [36] and rats [41]. Alauddin et al. reported that FRB could improve IR, decrease the serum leptin/adiponectin ratio, and activate the liver AMPK signaling pathway in stroke-prone spontaneously hypertensive rats [25]. In the present study, after the FRB intervention, a decreasing trend was shown in plasma leptin and an increasing trend was shown in the adiponectin/leptin ratio (Figure 6), which indicated that FRB might have the potential for attenuating the imbalanced adipokines if the feeding period was extended. On the other hand, small-molecule compounds found in FRB are given in Appendix A. The results indicated that the fermentation process might result in activation of some components such as ferulic acid and folic acid, which are involved in regulating lipid and glucose metabolism [42,43], demonstrating that fermentation may have the potential to release activated substances. Nevertheless, the present study indicated that it had no impact on the fasting glucose, insulin levels and HOMA-IRI among the groups (Figure 7). It was speculated that the therapeutic effect of FRB on NAFLD was inconsistent with past studies, which might have been due to the age of the animals, the fat composition in the HF diet, bacterial species used for fermentation, or the feeding period. Elucidating the exact mechanism requires further study. However, these results still showed that FRB may be beneficial for regulating lipid metabolism homeostasis in aged rats fed a HF diet.

### 3.4. Intestinal Damage

In a previous study, HF diet intake increased the intestinal permeability, decreased tight junction expressions, and activated liver Toll-like receptor 4 (TLR4)/nuclear factor (NF)-κB inflammation [44]. However, in this study, the HF diet did not change mRNA levels of intestinal tight junction proteins (Figure 8). The protein expressions of the intestinal tight junction protein should be determined in the future. According to the gut microbiota composition, compared to normal diet-fed rats, fecal samples from HF diet-fed rats were more enriched in Proteobacteria (phylum), Peptococcaceae (family), *rc4_4* (genus), *Dehalobacterium* (genus), and *Oscillospira* (genus) (Figure 12), which were similar to past studies in a HF diet-induced obesity rodent model [45,46]. At the species level, *Ruminococcus gnavus* was also abundant in the HF group, which was reported to alter the gut community in IBD patients [47].

On the other hand, the species diversity increased after supplementation with 1% FRB (Figure 9C), and some dominant bacteria, including the probiotic Lactobacillaceae (family) and SCFA-producing Lachnospiraceae (family), were observed in the FRB-treated groups (Figure 13 and Figure 14). A previous study revealed that FRB modulated the composition of the gut microbiota, especially enriching SCFA-producing bacteria such as *Dubosiella* and *Lactobacillus* and also increasing the SCFA levels in type 2 diabetic mice [35]. In a colitis mouse model, FRB enhanced SCFAs and tryptamine production, which promoted the tight junction barrier integrity [28]. Studies indicated that *Bacteroides uniformis* and *Bacteriodes acidifaciens*, which were overexpressed in the HF + 5% FRB group (Figure 11), could alleviate obesity-related metabolic dysfunction in mice [48,49]. In addition, the Coriobacteriaceae (family) was correlated with bile salts, steroid metabolism, and dietary polyphenol activation [50], which increased in the HF + 5% FRB group (Figure 14). Although some SCFA-producing bacteria were enriched in the FRB-treated groups, fecal SCFA levels were not significantly elevated in this study. The fecal propionic acid level only had an increasing trend after FRB supplementation (Table 7). The reason may be related to differences in the intervention dosage and periods compared with the previous studies. However, FRB still showed the potential to regulate the intestinal homeostasis by altering the composition of the gut microbiota in aged rats with HF diet feeding.

### 3.5. Gut-Liver Axis

Altered tight junction proteins may cause impairment of intestinal epithelial integrity, which would subsequently lead to bacteria translocation [51]. As mentioned previously, increased intestinal permeability and higher LPS levels were shown in NASH patients [51]. Changes in bacterial abundances in the gut microbiota could play a modulatory role in the gut-liver axis and pathogenesis of NAFLD [52]. In NAFLD patients, decreased bacterial diversity and lower levels of SCFA-producing members of the Lachnospiraceae, Lactobacillaceae, and Ruminococcaceae of the Firmicutes were observed [53]. In this study, rats fed a HF diet showed higher pathogenic bacterial abundances in fecal samples (intestinal damage), higher TNF-α levels and NAFLD score in the liver (liver injury), which might be the link between the intestinal and liver damage under HF diet feeding.

Dietary supplementation was thought to normalize the gut microbiome, which may be a beneficial strategy to regulate the NAFLD progression [54]. In this study, abundances of the beneficial Lachnospiraceae and Lactobacillaceae in the gut microbiota increased when HF diet-fed rats were supplemented with FRB (Figure 13 and Figure 14). Additionally, the hepatic IL-1β level and NAFLD score was inhibited in the FRB-treated groups (Table 3). As mentioned above, these results supported FRB supplementation possibly having a potential effect of maintaining a healthier gut microbiotic composition which was connected with the lower hepatic cytokine levels in rats fed the HF diet and FRB. On the other hand, the amino acids composition such as tryptophan, isoleucine and GABA (γ-Aminobutyric acid) of FRB were enhanced than that of RB (Appendix A). A previous study showed that tryptophan intervention supported intestinal integrity and ameliorated hepatic steatosis in NAFLD mouse model [55]. Therefore, higher level of functional amino acids in FRB seemed to play an important role in management of NAFLD by maintaining the gut and liver health.

### 3.6. Comparison between 1% and 5% of FRB Supplementation

Previous studies showed that 5% of FRB supplementation could attenuate metabolic syndrome [25] and obesity [35]. Muscle atrophy [26] and intestinal inflammatory disorders [28] were also prevented after 10% of RB fermented by *A. kawachii* and *Lactobacillus* sp. In the present study, rats in the HF+1% FRB and 5% FRB groups showed significantly lower hepatic IL-1β levels, a decreasing trend of final BWs, plasma leptin levels, and hepatic TNF-α levels compared to the HF group. Beneficial bacteria in the gut microbiota also increased in both HF+1% FRB and 5% FRB groups. Moreover, the NAFLD score was significantly reduced in the HF + 5% FRB group. Taken together, 5% of FRB treatment might have distinct effects in preventing liver damage, which is possibly due to the higher contents of nutrients and bioactive substances in FRB, including polyphenol compounds, ferulic acid and functional amino acids [56].

### 3.7. Study Limitations

This study still has several limitations. Hepatic protein expressions of leptin, adiponectin, lipogenesis-, and lipolysis-related pathways as well as intestinal tight junction proteins must be measured in order to clarify the protective effects of FRB on NAFLD in aged rats fed a HF diet through the mechanism of the gut-liver axis. Additionally, a young control group is required to discuss the relationship among age, HF diet consumption, and FRB supplementation. Lastly, extending the experimental period should be considered in future studies.

## 4. Materials and Methods

### 4.1. Animals

Male Sprague Dawley (SD) rats at 36 weeks of age (BioLasco Taiwan, Ilan, Taiwan) were used in the present study. All rats were housed in individual cages in an animal room maintained at 23 ± 2 °C with 55% ± 10% relative humidity and a 12 h light–dark cycle. All rats were allowed free access to a standard rodent diet (LabDiet 5001 Rodent Diet; PMI Nutrition International, St. Louis, MO, USA) and water for 8 weeks of acclimation before the study. All procedures were approved by the Institutional Animal Care and Use Committee of Taipei Medical University (with an identification code of LAC-2020-0143).

### 4.2. Study Protocol

After 8 weeks of acclimatization, all rats were randomly divided into four groups (six rats/group) based on their BWs, including a normal control (NC) group (normal diet), high-fat (HF) group (HF diet; 60% of total calories from fat), HF + 1% FRB group (HF diet + 1%FRB *w/w*), and HF + 5% FRB group (HF diet + 5%FRB *w/w*). The diet formula was designed according to the method of Islam et al. [28]. Details of the experimental diets are given in Table 8. All rats were fed their diet and water *ad libitum*, and the BW change and diet consumption were measured every week. The experiment was carried out for 8 weeks. At the end of the 7th week of the study, feces were collected for fecal SCFA and microbiotic analyses. After 8 weeks of the experimental period, rats were anesthetized and sacrificed. Blood samples were collected in heparin-containing tubes and centrifuged (3000 rpm for 20 min at 4 °C) to obtain plasma. All of the plasma samples were stored at −80 °C until being assayed. Liver and intestinal tissues were rapidly excised and stored at −80 °C for further analysis.

### 4.3. Fermentation Processes and Analysis of FRB Components

#### 4.3.1. Production Procedures of FRB

Rice bran was fermented with *Aspergillus kawachii* (Bioresource Collection and Research Center, Hsinchu, Taiwan), and FRB was produced based on procedures described in a previous study [25]. Rice bran (100g; YuanShun, Yunlin, Taiwan) and 100 g of fat-free rice bran (Guanshan, Taitung, Taiwan) were initially mixed in a stainless-steel container and steamed for 15 min, followed by the addition of 50 g of purified water to mix the ingredients well. Then, the mixture was sterilized (1.2 atm for 30 min at 121 °C) and cooled to about 30 °C, after which a spore solution of *A. kawachii* was inoculated and placed in a microcentrifuge tube containing 1 mL of distilled water (dH_2_O) to wash out the spores until the solution turned light-proof black. The spore solution was then diluted with 1000 mL of dH_2_O and evenly shaken. The spore solution was dropped into the culture medium and incubated at 25 °C in a fermentation chamber overnight, then the water was poured from the medium, and then poured again once at 08:00 and 16:00. After 4 days of fermentation, the FRB product was obtained. The FRB solution was stirred with equal proportions of purified water and centrifuged (3000 rpm for 15 min at 4 °C), condensed for 30~40 min, and lyophilized at −40 to −60 °C for 3 days to collect FRB extracts. The lyophilized powder was kept at 4 °C until being used.

#### 4.3.2. Antioxidative Status of FRB

##### Total Antioxidant Capacity (TAC)

The TAC of FRB was determined according to the manufacturer’s instruction (Cayman Chemical, Ann Arbor, MI, USA). The optical density (OD) was read at 750 nm with a microplate reader (Molecular Devices, Sunnyvale, CA, USA).

##### Di(Phenyl)-(2,4,6-trinitrophenyl) Iminoazanium (DPPH) Antioxidant Assay

Radical-scavenging activity was quantified by a DPPH antioxidant assay using a commercial kit (D678-01, Dojindo, Rockville, MD, USA) and measuring the absorbance at 517 nm.

### 4.4. Measurements and Analytical Procedures

#### 4.4.1. Determination of Liver Damage

##### Liver Function Index

Activities of plasma aspartate aminotransferase (AST) and alanine aminotransferase (ALT) were detected with the ADVIA^®^ Chemistry XPT System (Siemens Healthcare Diagnostics, Eschborn, Germany).

##### Liver Histopathological Examination

The caudate lobe of liver tissues was fixed in a 10% formaldehyde solution. Sections of liver tissues were then stained with hematoxylin A- and eosin (H&E) and evaluated by a veterinarian. Images of the tissues were captured with a digital camera at 200× magnification. Histopathological evaluations of macrovesicular steatosis, microvesicular steatosis, hypertrophy, and the number of inflammatory foci were separately scored, and the severity was graded as described by Liang et al. [57] with minor modifications. Adding the scores of the above four parameters was used to calculate the NAFLD score.

##### Liver Cytokine Levels

The extraction method of liver tissues was described by Chen et al. [58]: 0.5 g of liver tissues was homogenized in 1.5 mL of ice-cold buffer containing 50 mM Tris (pH 7.2), 150 mM NaCl, 1% Triton X-100, and 0.1% protease inhibitor (PI) (HYK0010, MedChemExpress, Monmouth Junction, NJ, USA). The homogenized solution was then centrifuged at 3000 rpm for 15 min at 4 °C, and the supernatant was collected. Concentrations of hepatic TNF-α, IL-1β, IL-6, and IL-10 were determined by corresponding enzyme-linked immunosorbent assay (ELISA) kits, including rat TNF-α (BioLegend Systems, San Diego, CA, USA), IL-1β (Rat IL-1, R&D Systems, Minneapolis, MN, USA), IL-6 (Rat IL-6, R&D Systems) and IL-10 (Rat IL-10, R&D Systems, Minneapolis, MN, USA). The OD was read at 450 nm with a microplate reader (Molecular Devices, Sunnyvale, CA, USA) for all cytokines.

##### Plasma and Liver Lipid Peroxidation

Liver tissues (0.1 g) were homogenized as described for the method of liver cytokines. Plasma and the supernatants of liver homogenates were used for the lipid peroxidation analysis. Lipid peroxidation in plasma and the liver was measured by TBARS levels with a TBARS kit (TBARS 10009055 (TCA method) Assay Kit, Cayman Chemical, MI, USA) according to the assay kit instructions.

#### 4.4.2. Determination of Lipid Metabolism

##### Hepatic TC and TG Concentrations

For the liver TC determination, 0.01 g of liver tissue was homogenized in 200 μL of solvent (chloroform: isopropanol: nonyl phenoxypolyethoxylethanol (NP-40) of 7: 11: 0.1) and centrifuged at 8876 rpm for 10 min at 4 °C. Avoiding the pellet, the liquid (organic phase) was transferred to a new tube, air dried at 50 °C to remove the chloroform, and samples were placed under a vacuum for 30 min to remove trace amounts of the organic solvent. Dried lipids were then dissolved in 200 μL of 1× assay diluent with vortexing until the solution turned homogenous and then was stored at −80 °C. To measure the concentrations of liver TGs, 0.1 g of liver tissue was homogenized in 500 μL of diluted NP-40 reagent containing protease inhibitors. The mixture was centrifuged at 7247 rpm for 10 min at 4 °C, and the supernatant (including a layer of insoluble fat) was collected and stored at −80 °C. TC and TG contents were further analyzed with a cholesterol colorimetric assay kit (Cell Biolabs, San Diego, CA, USA) and colorimetric TG assay kit (Cayman Chemical), and results were expressed as milligrams per gram (mg/g) of liver tissue.

##### Hepatic Lipid Metabolism-Related Protein Expressions

A Western blot analysis was performed to determine expressions of SIRT1, AMPKα, p-AMPKα, leptin receptor and AdipoR2. The method of crude extraction preparation of liver tissues was assessed according to previously described protocols [59]. Liver tissue (0.1 g) was homogenized in 0.5 mL of RIPA buffer containing 1% of a PI (HYK0010, MedChemExpress) and phosphatase inhibitor (PPI) (HYK0022, MedChemExpress). After being placed in an ice bath for 30 min, the homogenate was centrifuged at 7939 rpm for 15 min at 4 °C, followed by collection of the supernatant. Liver proteins (50 μg) were separated by 10% sodium dodecylsulfate polyacrylamide gel electrophoresis (SDS-PAGE), and then electroblotted onto polyvinylidene difluoride (PVDF) transfer membranes (Pall, Port Washington, NY, USA) and incubated with 5% bovine serum albumin (BSA). These blots were incubated with primary antibodies listed in Table 9, and glyceraldehyde 3-phosphate dehydrogenase (GAPDH) was used as an internal control. The blots were finally treated with goat anti-rabbit immunoglobulin G (IgG) horseradish peroxidase (HRP; Croyez Bioscience, Taipei, Taiwan), and specific bindings of anti-bodies were assayed by the UVP Chemidoc it 515 Imaging System (UVP, Upland, CA, USA) using a T Western Lightning kit (PerkinElmer Lifesciences, Boston, MA, USA). Bands were quantified using Image-Pro Plus software (Media Cybernetics, Rockville, MD, USA).

##### Hepatic FA Metabolism-Related Gene mRNA Levels

Total RNA of the liver was extracted with the TRI Reagent^®^ (Sigma-Aldrich, St. Louis, MO, USA), according to instructions from the manufacturer. The quality and quantity of total RNA were evaluated by measuring the OD 260/280 ratio on a BioTek epoch reader with the Gen5TM Take3 Module (BioTek Instruments, Winooski, VT, USA). The concentration of total RNA was adjusted to 4000 ng/µL and then reverse-transcribed with a RevertAid First Strand cDNA Synthesis kit (#K1621, ThermoFisher Scientific, Waltham, MA, USA). The concentration of complementary (c)DNA was calculated by the BioTek epoch reader with the Gen5TM Take3 Module system and adjusted to 50 ng/µL. The resulting cDNA was amplified in a 96-well polymerase chain reaction (PCR) plate using SYBR Green/ROX qPCR Master Mix (2X) (ThermoFisher Scientific) on a QuantStudio 1 Real-Time PCR System (ThermoFisher Scientific). Gene levels were normalized to β-actin, and the ratio to β-actin was calculated by setting the value of the NC group to 1. Information on primers is given in Table 10.

##### Plasma Adipokine Levels

A Leptin Mouse/Rat ELISA kit (Biovendor, Brno, Czech Republic) and Rat Adiponectin ELISA kit (Assaypro, Charles, MO, USA) were used to detect plasma leptin and adiponectin levels, respectively, following the manufacturer’s instructions. The absorbance was read on a microplate reader (Molecular Devices) at 450 nm for all adipokines.

#### 4.4.3. IR Analysis

Plasma insulin levels were assayed with a Rat Insulin ELISA kit (Mercodia, Uppsala, Sweden). The fasting blood glucose level was detected via the Glucose Monitoring System (Abbott Diabetes Care, Oyl, UK). The HOMA-IRI was calculated with the following formula: HOMA-IRI = fasting blood glucose (mg/dL) × fasting insulin (mIU/L)/405.

#### 4.4.4. Determination of Intestinal Damage

##### Intestinal Tight Junction Protein mRNA Levels

The intestinal tight junction protein mRNA levels were measured by quantitative (q)PCR methods. The method of ileum sample preparation was the same as that for the measurement of hepatic FA metabolism-related genes. Information on primers is given in Table 10.

##### Fecal Microbiotic Analysis

The fecal microbiotic composition was analyzed using a 16S ribosomal (r)RNA Next Generation Sequencing (NGS) system. Fresh feces were collected into sterilized 2-mL Eppendorf tubes and stored at −80 °C for analysis. The amplified fecal DNA was purified with Agencourt AMPure XP Reagent beads (Beckman Coulter, Brea, CA, USA). A qPCR (KAPA SYBR FAST qPCR Master Mix) was used to quantify each library with the Roche LightCycler 480 system, and then pooled equally to 4 nM for the Illumina MiSeq NGS system (illumina, San Diego, CA, USA). More than 80,000 reads with paired-end sequencing (>250 bp*2) were generated, and the QIIME2 workflow [60] classified organisms from the amplicons using the GreenGenes taxonomy database. MicrobiomeAnalyst [61] was used to perform statistical and visual analyses of the microbiome data.

##### Fecal SCFA Concentrations

Fecal SCFAs were extracted according to the method described by García-Villalba et al. [62] with slight modification. Fresh feces were weighed and suspended in 1 mL of water with 0.5% phosphoric acid per 0.1 g of sample into sterilized 2-mL Eppendorf tubes and stored at −80 °C until analysis. The SCFA analysis was performed by gas chromatographic-mass spectrometric (GC-MS) system of Agilent 5977B coupled with a 7820A autoinjector (Agilent Technologies, Santa Clara, CA, USA). For GC-MS, a Nukol™ 30 m × 0.25 × 0.25 µm capillary column (24107, Supelco, Bellefonte, PA, USA) was used. Injection was made in the pulsed split mode with an injection volume of 1 µL and an injector temperature of 250 °C. GC separation was carried out at 90 °C initially, then increased to 150 °C at 15 °C /min, to 170 °C at 3 °C/min and finally to 200 °C at 50 °C/min (total time 16.267 min). The solvent delay was 3.2 min. The identification of SCFAs was based on the retention times of standard compounds using a commercial standard solution (46975-U, Supelco). Data were quantitated with MassHunter Quantitative Analysis software (Agilent Technologies).

#### 4.4.5. Statistical Analysis

Data are expressed as the mean ± standard deviation (SD). Statistical analysis was performed using GraphPad Prism vers. 8.0.1 (GraphPad Software, San Diego, CA, USA). Student’s *t*-test was used to determine statistical differences between the NC and HF groups. A one-way analysis of variance (ANOVA), followed by Fisher’s post hoc test was used to determine statistical differences among the HF, HF + 1% FRB, and HF + 5% FRB groups. Statistical significance was assigned at the *p* < 0.05 level.

## 5. Conclusions

It was found that the water extract of fermented rice bran (by *Aspergillus kawachii*) had a higher anti-oxidative ability. The current results demonstrated that after 8 weeks of FRB treatment, rats in the HF + 5% FRB group had a significantly lower NAFLD score and hepatic IL-1β level, a decreasing trend of final BW and plasma leptin level, as well as an increase in beneficial bacteria in the gut microbiota. In summary, it was suggested that FRB showed the potential for alleviating liver damage induced by a HF diet, possibly through regulating the imbalanced adipokines and altering the gut microbiotic composition (Figure 15).

## Figures and Tables

**Figure 1 plants-11-00607-f001:**
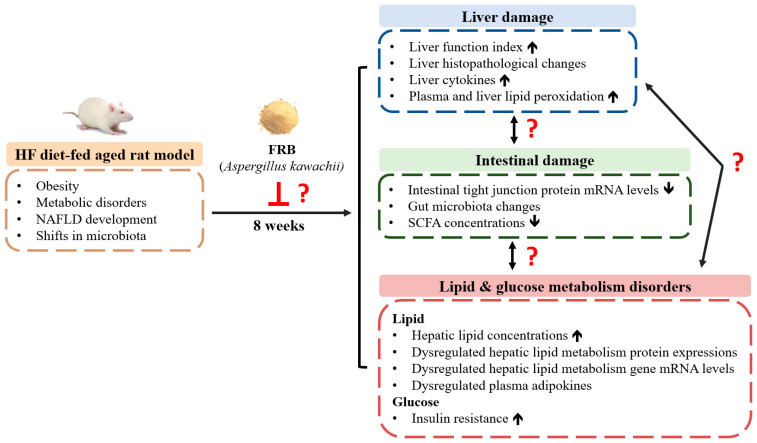
The research motivation and specific aims. NAFLD, non-alcoholic fatty liver disease; HF, high-fat; FRB, water extract of fermented rice bran; SCFA, short-chain fatty acid.

**Figure 2 plants-11-00607-f002:**
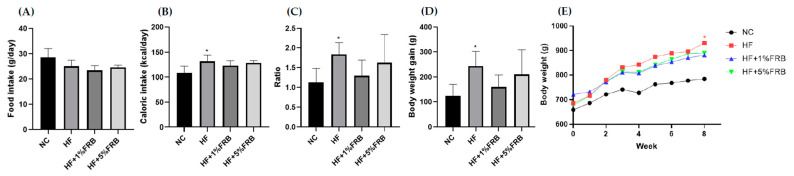
Effects of the water extract of fermented rice bran (FRB) on food intake and body weight in aged rats with high-fat (HF) diet feeding. (**A**) Food intake, (**B**) caloric intake, (**C**) food efficiency ratio (FER), (**D**) body weight gain and (**E**) body weight changes over time. The FER was calculated by applying the equation: FER = (body weight gain (g)/food intake (g)). Values are presented as the mean ± standard deviation (*n* = 6). * *p* < 0.05 vs. the normal control (NC) group; significance between two groups was determined using Student’s *t*-test. In the HF diet-fed groups, significant differences between groups were determined by a one-way ANOVA with Fisher‘s post hoc test.

**Figure 3 plants-11-00607-f003:**
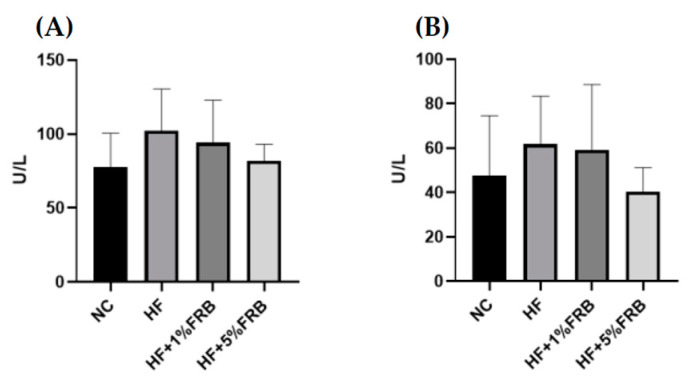
Effects of water extract of fermented rice bran (FRB) on liver function index in aged rats with high-fat (HF) diet feeding. Plasma level of (**A**) aspartate aminotransferase (AST) and (**B**) alanine aminotransferase (ALT). Values are presented as the mean ± standard deviation (*n* = 6). * *p* < 0.05 vs. the normal control (NC) group; significance between two groups was determined using Student’s *t*-test. In the HF diet-fed groups, differences between groups were determined by a one-way ANOVA with Fisher‘s post hoc test.

**Figure 4 plants-11-00607-f004:**
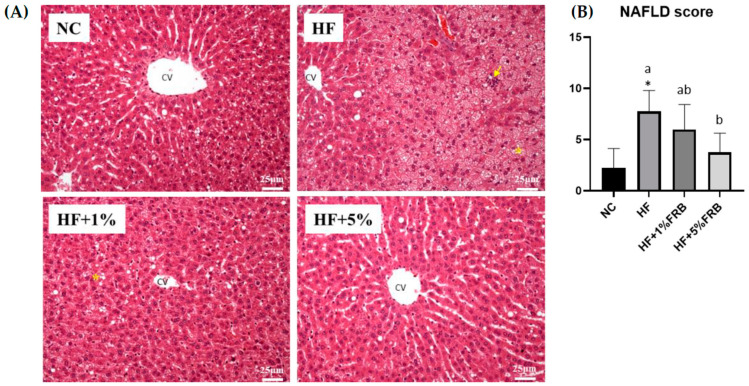
Effects of the water extract of fermented rice bran (FRB) on liver histopathological changes in aged rats with high-fat (HF) diet feeding. (**A**) H&E staining. Magnification: ×200. Scale bar, 25 μm. (Yellow arrow: inflammatory cell infiltration; yellow star: lipid droplets.) (**B**) Histopathological analysis scores. Values are presented as the mean ± standard deviation (*n* = 4). * *p* < 0.05 vs. the normal control (NC) group; significance between two groups was determined using Student’s *t*-test. In the HF diet-fed groups, different letters indicate significant differences between groups at *p* < 0.05 by a one-way ANOVA with Fisher‘s post hoc test. NAFLD, nonalcoholic fatty liver disease.

**Figure 5 plants-11-00607-f005:**
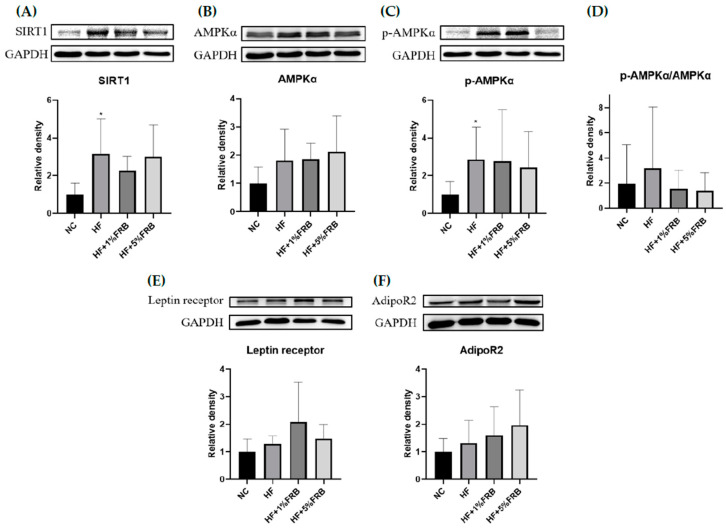
Effects of the water extract of fermented rice bran (FRB) on hepatic lipid metabolism-related protein expressions in aged rats with high-fat (HF) diet feeding. Values are presented as the mean ± standard deviation (*n* = 6). * *p* < 0.05 vs. the normal control (NC) group; significance between two groups was determined using Student’s *t*-test. In the HF diet-fed groups, significant differences between groups were determined by a one-way ANOVA with Fisher‘s post hoc test. Western blot analysis of (**A**) NAD-dependent sirtuin-1 (SIRT1), (**B**) adenosine monophosphate-activated protein kinase-α (AMPKα), (**C**) phosphorylated (p)-AMPKα, (**D**) p-AMPKα/AMPKα, (**E**) leptin receptor, and (**F**) adiponectin receptor 2 (AdipoR2) protein expressions. Glyceraldehyde 3 phosphate dehydrogenase (GAPDH) was used as an internal control. Quantitative analysis of protein levels and the ratio of each internal control were calculated by setting the value of the mean of the NC group.

**Figure 6 plants-11-00607-f006:**
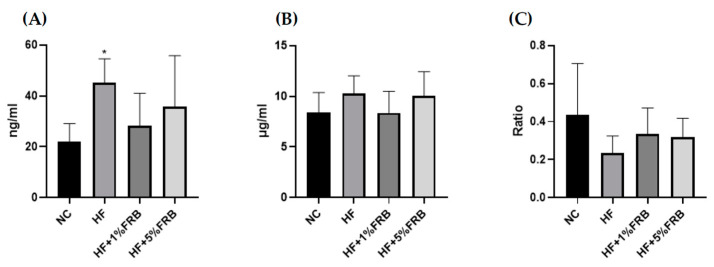
Effects of the water extract of fermented rice bran (FRB) on plasma adipokine levels in aged rats with high-fat (HF) diet feeding. Plasma levels of (**A**) leptin, (**B**) adiponectin, and (**C**) the adiponectin/leptin ratio. Values are presented as the mean ± standard deviation (*n* = 6). * *p* < 0.05 vs. the normal control (NC) group; significance between two groups was determined using Student’s *t*-test. In the HF diet-fed groups, significant differences between groups were determined by a one-way ANOVA with Fisher‘s post hoc test.

**Figure 7 plants-11-00607-f007:**
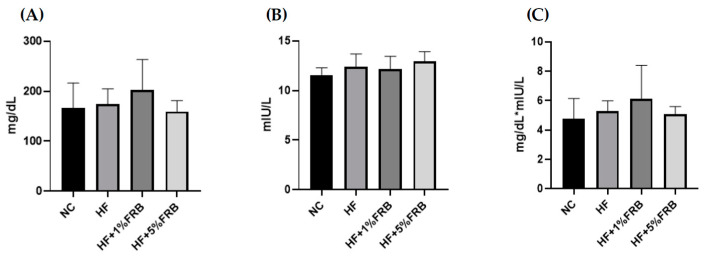
Effects of the water extract of fermented rice bran (FRB) on the insulin resistance analysis in aged rats with high-fat (HF) diet feeding. (**A**) Fasting blood glucose level, (**B**) fasting plasma insulin level, and (**C**) homeostasis model assessment of the insulin resistance index (HOMA-IRI). Values are presented as the mean ± standard deviation (*n* = 6). Significance between the normal control (NC) and HF groups was determined using Student’s *t*-test. In the HF diet-fed groups, significant differences between groups were determined by a one-way ANOVA with Fisher‘s post hoc test.

**Figure 8 plants-11-00607-f008:**
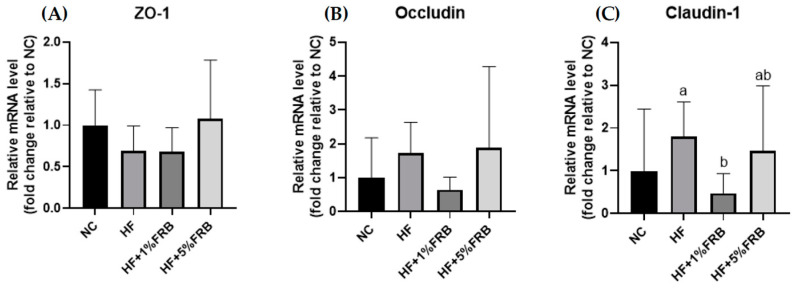
Effects of the water extract of fermented rice bran (FRB) on intestinal zonula occludens (ZO)-1, occludin, and claudin-1 mRNA levels in aged rats with high-fat (HF) diet feeding. Values are presented as the mean ± standard deviation (*n* = 6). * *p* < 0.05 vs. the normal control (NC) group; significance between two groups was determined using Student’s *t*-test. In the HF diet-fed groups, different letters indicate significant differences between groups at *p* < 0.05 by a one-way ANOVA with Fisher’s post hoc test. Comparative quantification of each gene was normalized to β-actin using the 2^−∆∆Ct^ method, and the ratio of each internal control was calculated by setting the value of the mean of the NC group.

**Figure 9 plants-11-00607-f009:**
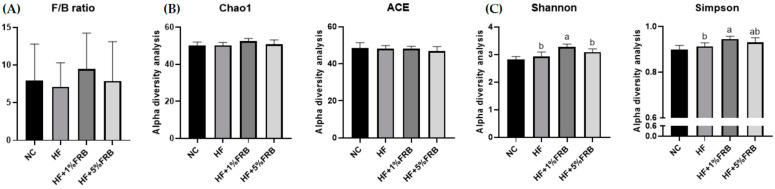
Effects of the water extract of fermented rice bran (FRB) on the Firmicutes-to-Bacteroidetes (F/B) ratio and α-diversity of the fecal microbiota in aged rats with high-fat (HF) diet feeding. (**A**) F/B ratio. (**B**) Community richness of the fecal microbiota. (**C**) Community diversity of the fecal microbiota. Values are presented as the mean ± standard deviation (*n* = 5). * *p* < 0.05 vs. the normal control (NC) group; significance between two groups was determined using Student’s *t*-test. In the HF diet-fed groups, different letters indicate significant differences between groups at *p* < 0.05 by a one-way ANOVA with Fisher’s post hoc test.

**Figure 10 plants-11-00607-f010:**
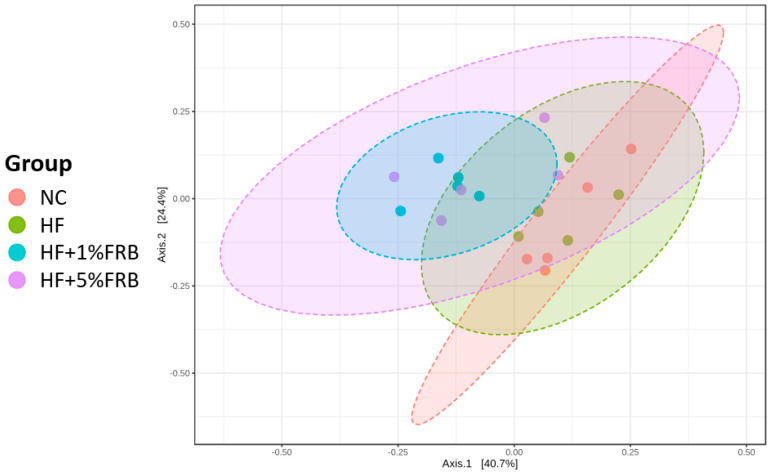
Effects of the water extract of fermented rice bran (FRB) on a principal coordinate analysis (PCoA) of the fecal microbiota in aged rats with high-fat (HF) diet feeding. Values are presented as the mean ± standard deviation (*n* = 5).

**Figure 11 plants-11-00607-f011:**
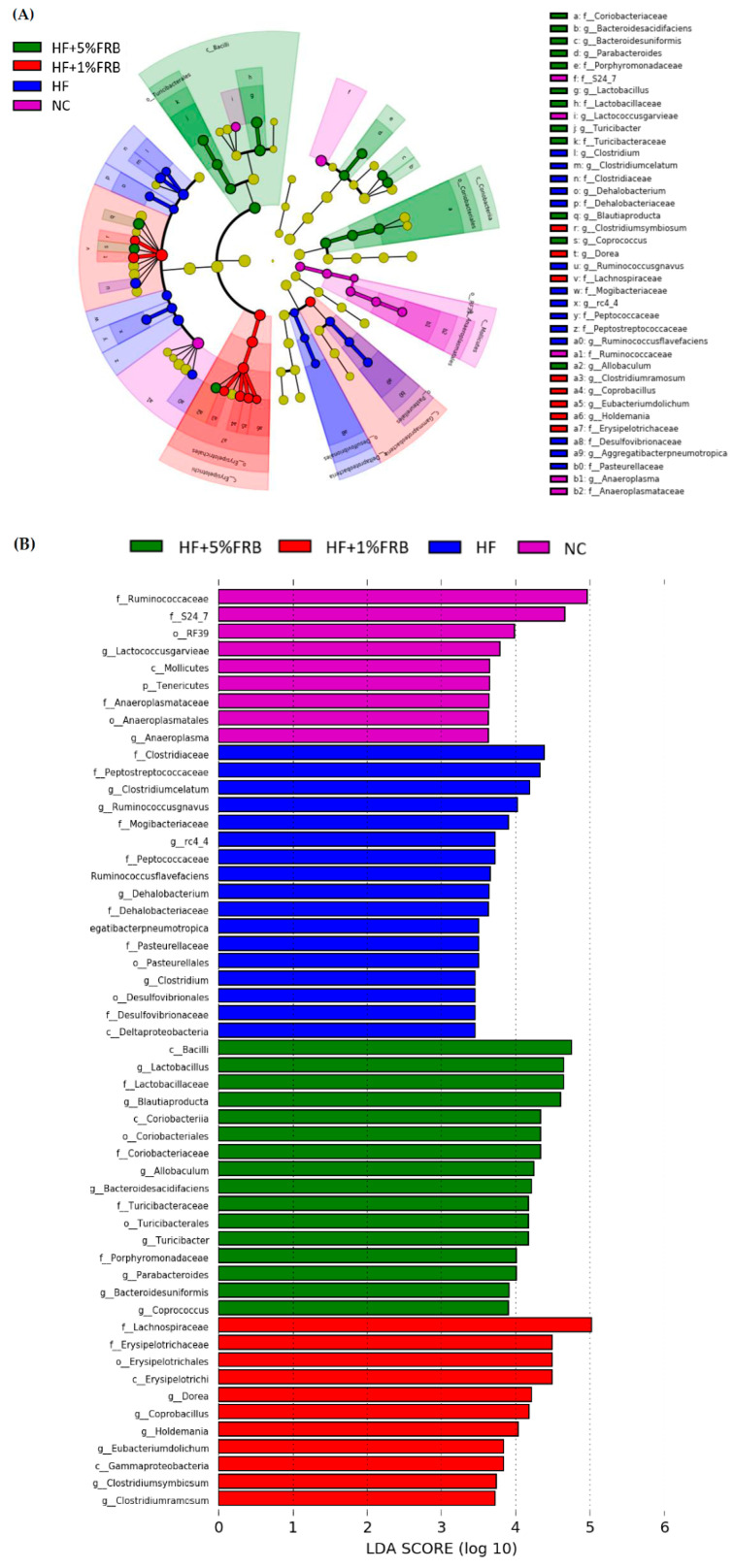
Effects of the water extract of fermented rice bran (FRB) on taxonomies of fecal microbiotic compositions in aged rats with high-fat (HF) diet feeding. (**A**) A linear discriminant analysis of the effect size (LEfSe) of the most significant abundance differences in the fecal microbiota among all groups (*n* = 5). (**B**) Bacteria meeting the LDA threshold (≥2) differed among all groups (*n* = 5).

**Figure 12 plants-11-00607-f012:**
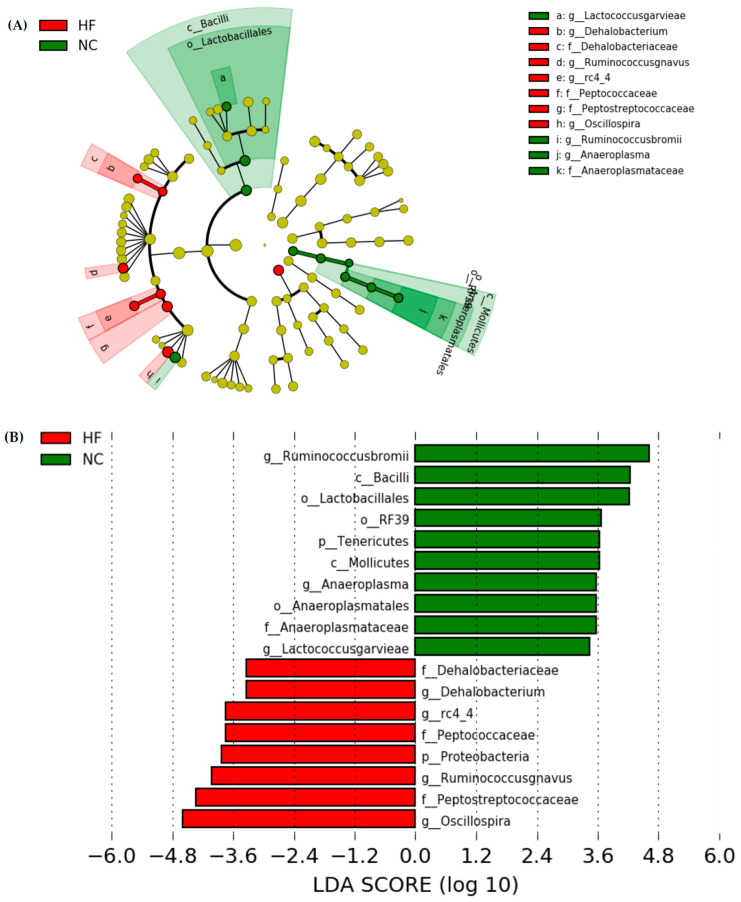
Effects of the water extract of fermented rice bran (FRB) on taxonomies of fecal microbiotic compositions in aged rats with high-fat (HF) diet feeding. (**A**) A linear discriminant analysis of the effect size (LEfSe) of the most significant abundance differences in the fecal microbiota in the NC and HF groups (*n* = 5). (**B**) Bacteria meeting the LDA threshold (≥2) differed in the NC and HF groups (*n* = 5).

**Figure 13 plants-11-00607-f013:**
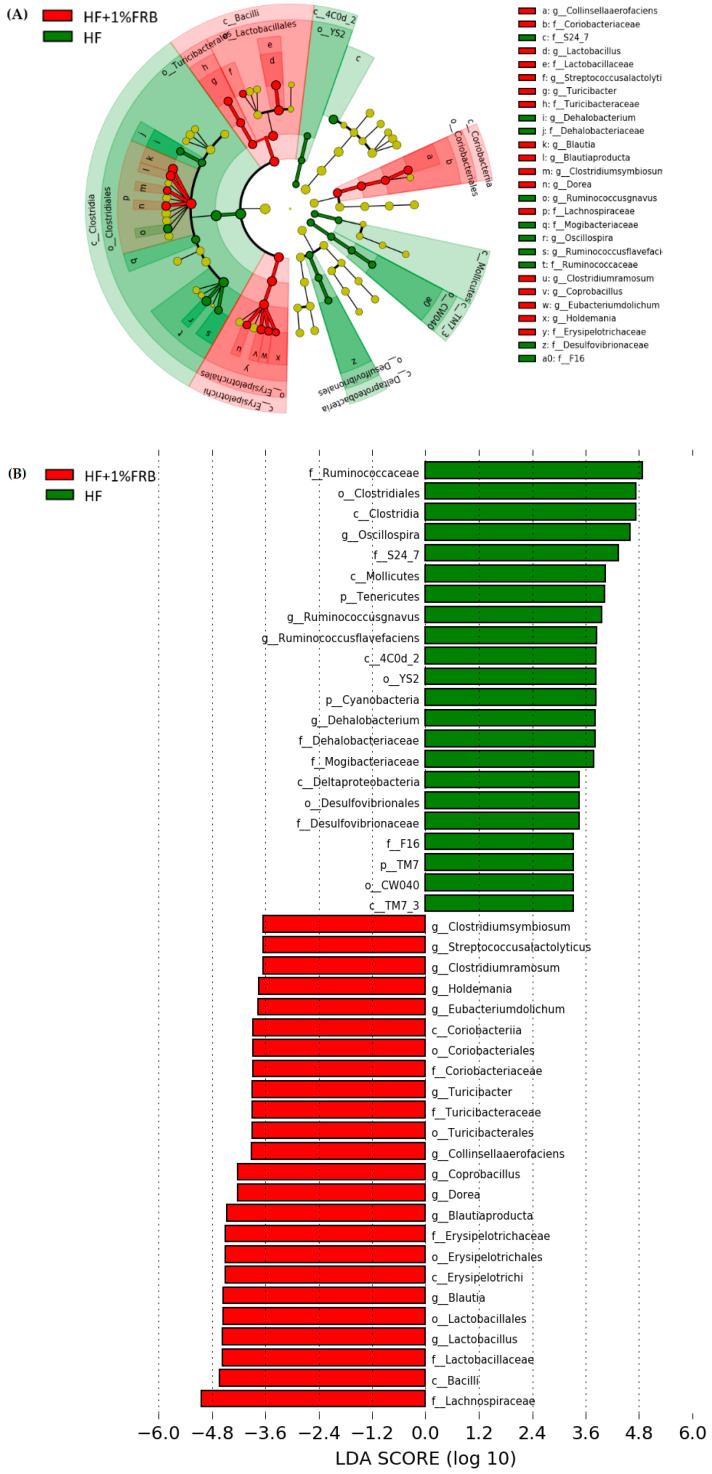
Effects of the water extract of fermented rice bran (FRB) on taxonomies of fecal microbiotic compositions in aged rats with high-fat (HF) diet feeding. (**A**) A linear discriminant analysis of the effect size (LEfSe) of the most significant abundance differences in the fecal microbiota in the HF and HF + 1% FRB groups (*n* = 5). (**B**) Bacteria meeting the LDA threshold (≥2) differed in the HF and HF + 1% FRB groups (*n* = 5).

**Figure 14 plants-11-00607-f014:**
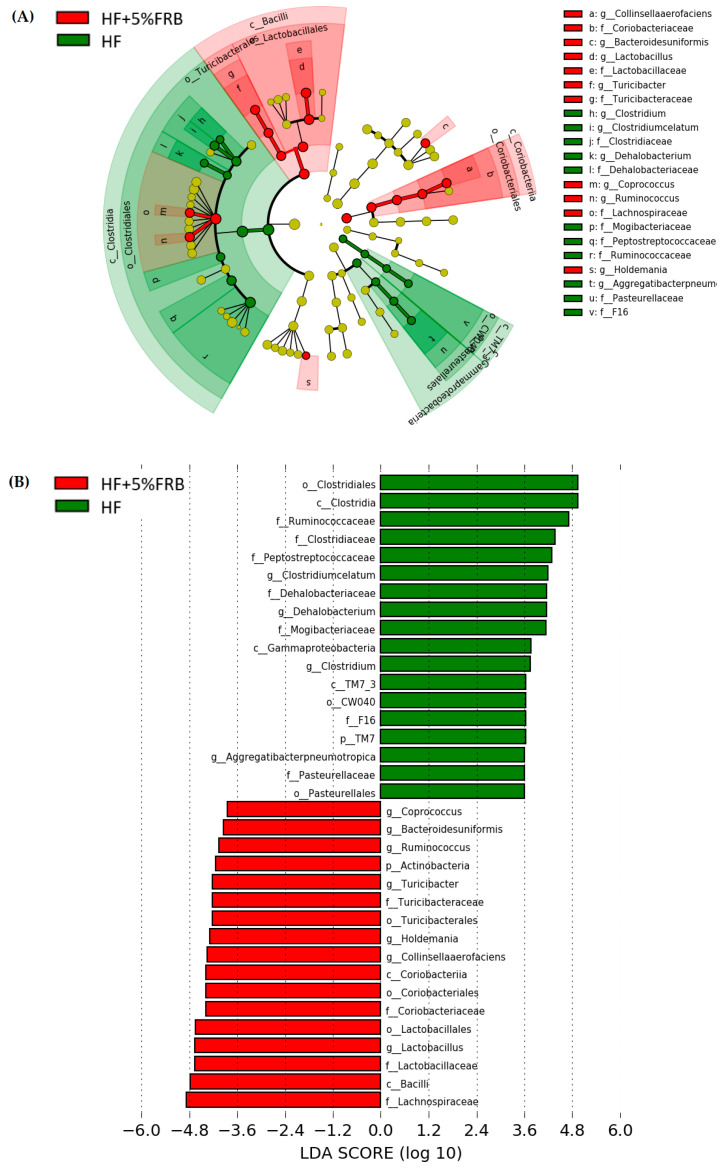
Effects of the water extract of fermented rice bran (FRB) on taxonomies of fecal microbiotic compositions in aged rats with high-fat (HF) diet feeding. (**A**) A linear discriminant analysis of the effect size (LEfSe) of the most significant abundance differences in the fecal microbiota in the HF and HF + 5% FRB groups (*n* = 5). (**B**) Bacteria meeting the LDA threshold (≥2) differed in the HF and HF + 5% FRB groups (*n* = 5).

**Figure 15 plants-11-00607-f015:**
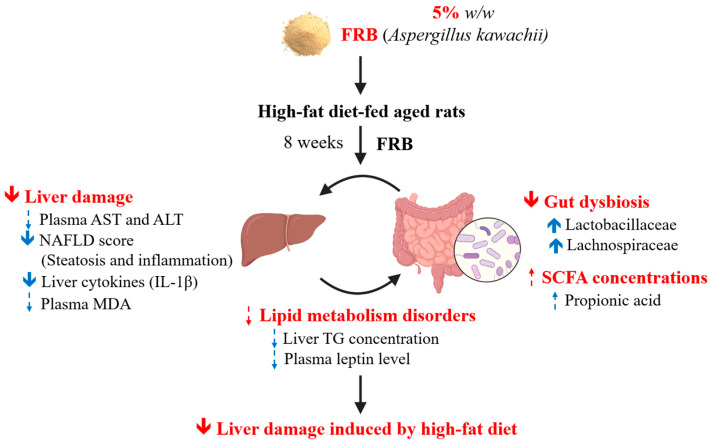
Effects of the water extract of fermented rice bran (FRB) on liver damage and intestinal injury in aged rats with high-fat (HF) diet feeding. In this study, it was indicated that FRB ameliorated liver damage induced by the HF diet which was represented as a lower non-alcoholic fatty liver disease (NAFLD) score and hepatic interleukin (IL)-1β level in rats. The protective effects of FRB against liver damage may have been due to regulating the plasma adipokines and maintaining homeostasis of the gut microbiota. Solid blue arrow: levels of analytical items were significantly increased or decreased in the HF + 5% FRB group than in the HF group; dotted blue arrow: levels of analytical items in the HF + 5% FRB group showed an increasing or decreasing trend compared to the HF group.

**Table 1 plants-11-00607-t001:** Antioxidant ability of the water extract of fermented rice bran (RB; FRB).

	RB	FRB
Total antioxidant capacity (mM)	65.38 ± 2.49	77.38 ± 0.27
Inhibition ratio (% per 100 mg)	9.05	64.10
Trolox equivalent (per 100 mg)	1578.13	10,096.88

All values are presented as the mean ± standard deviation. RB, water extract of rice bran; FRB, water extract of fermented rice bran.

**Table 2 plants-11-00607-t002:** Histopathological analysis scores.

Item	Definition	Score	Groups			
			NC	HF	HF + 1% FRB	HF + 5% FRB
Macrovesicular steatosis	<5%	0	0.75 ± 0.96	0.25 ± 0.50	0.25 ± 0.50	0.00 ± 0.00
	5~33%	1				
	33~66%	2				
	>66%	3				
Microvesicular steatosis	<5%	0	0.75 ± 0.50	1.75 ± 0.50 *	1.25 ± 0.50	1.00 ± 0.82
	5~33%	1				
	33~66%	2				
	>66%	3				
Hepatocellular hypertrophy	<5%	0	0.50 ± 0.58	2.50 ± 0.58 *	1.75 ± 0.96	2.25 ± 0.96
	5~33%	1				
	33~66%	2				
	>66%	3				
Number of inflammatory foci	0~2	0	0.25 ± 0.50	3.25 ± 0.96 *^,a^	2.75 ± 1.26 ^a^	0.50 ± 0.58 ^b^
	3~5	1				
	6~9	2				
	10~19	3				
	>20	4				
NAFLD activity score	Sum of score	0~13	2.25 ± 1.89	7.75 ± 2.06 *^,a^	6.00 ± 2.45 ^a,b^	3.75 ± 1.89 ^b^

Values are presented as the mean ± standard deviation (*n* = 4). * *p* < 0.05 vs. the normal control (NC) group; significance between two groups was determined using Student’s *t*-test. In the high-fat (HF) diet fed groups, different letters indicate significant differences between groups at *p* < 0.05 by a one-way ANOVA with Fisher’s post hoc test. NAFLD, nonalcoholic fatty liver disease; FRB, water extract of fermented rice bran.

**Table 3 plants-11-00607-t003:** Effects of the water extract of fermented rice bran (FRB) on liver cytokine levels in aged rats with high-fat (HF) diet feeding.

pg/mg Protein	NC	HF	HF + 1% FRB	HF + 5% FRB
TNF-α	7.49 ± 1.40	9.25 ± 0.54 *	8.25 ± 1.32	8.45 ± 1.24
IL-1β	46.21 ± 9.01	55.27 ± 7.68 ^a^	44.03 ± 5.11 ^b^	35.53 ± 4.64 ^c^
IL-6	7.92 ± 2.80	8.24 ± 0.43	8.72 ± 2.10	8.82 ± 2.67
IL-10	2.54 ± 0.55	1.95 ± 0.37	1.49 ± 0.29	1.46 ± 0.61

Values are presented as the mean ± standard deviation (*n* = 6). * *p* < 0.05 vs. the normal control (NC) group; significance between two groups was determined using Student’s *t*-test. In the HF diet-fed groups, different letters indicate significant differences between groups at *p* < 0.05 by a one-way ANOVA and Fisher’s post hoc test. TNF, tumor necrosis factor; IL, interleukin.

**Table 4 plants-11-00607-t004:** Effects of the water extract of fermented rice bran (FRB) on plasma and liver thiobarbituric acid-reactive substances (TBARSs) in aged rats with high-fat (HF) diet feeding.

	NC	HF	HF + 1% FRB	HF + 5% FRB
Plasma TBARS (ng/μL)	2.51 ± 0.98	4.85 ± 2.67	4.26 ± 2.81	3.07 ± 0.87
Liver TBARS (ng/mg)	44.04 ± 21.57	42.34 ± 9.54	59.43 ± 23.16	47.59 ± 32.50

Values are presented as the mean ± standard deviation (*n* = 6). Significance between the normal control (NC) and HF groups was determined using Student’s *t*-test. In the HF diet-fed groups, significant differences between groups at *p* < 0.05 by a one-way ANOVA with Fisher’s post hoc test.

**Table 5 plants-11-00607-t005:** Effects of the water extract of fermented rice bran (FRB) on hepatic total cholesterol (TC) and triglyceride (TG) levels in aged rats with high-fat (HF) diet feeding.

mg/g	NC	HF	HF + 1% FRB	HF + 5% FRB
Liver TC	3.82 ± 1.34	3.50 ± 0.52	3.48 ± 1.58	3.10 ± 0.68
Liver TG	60.68 ± 23.64	71.22 ± 15.26	67.30 ± 20.06	59.43 ± 4.57

Values are presented as the mean ± standard deviation (*n* = 6). Significance between the normal control (NC) and HF groups was determined using Student’s *t*-test. In the HF diet-fed groups, significant differences between groups were determined by a one-way ANOVA with Fisher‘s post hoc test.

**Table 6 plants-11-00607-t006:** Effects of the water extract of fermented rice bran (FRB) on hepatic fatty acid metabolism-related gene mRNA levels in aged rats with high-fat (HF) diet feeding.

mRNA Levels	NC	HF	HF + 1% FRB	HF + 5% FRB
SREBP-1c	1.00 ± 0.56	1.04 ± 0.85	0.85 ± 0.66	0.59 ± 0.27
ACC	1.00 ± 0.36	1.61 ± 1.32	1.43 ± 1.10	1.03 ± 0.49
SCD1	1.00 ± 0.22	0.59 ± 0.53	0.39 ± 0.31	0.21 ± 0.16
FAS	1.00 ± 0.51	0.69 ± 0.76	0.28 ± 0.19	0.24 ± 0.12
PPARα	1.00 ± 0.42	0.86 ± 0.20	0.57 ± 0.41	0.88 ± 0.64
CPT-1	1.00 ± 0.59	0.84 ± 0.15 ^a^	0.45 ± 0.27 ^b^	0.69 ± 0.36 ^a,b^

Values are presented as the mean ± standard deviation (*n* = 6). Significance between the normal control (NC) and HF groups was determined using Student’s *t*-test. In the HF diet-fed groups, different letters indicate significant differences between groups at *p* < 0.05 by a one-way ANOVA with Fisher’s post hoc test. Comparative quantification of each gene was normalized to β-actin using the 2^−∆∆Ct^ method and the ratio of each internal control was calculated by setting the value of the mean of the NC group. SREBP-1c, sterol response element-binding protein-1c; ACC, acetyl CoA carboxylase; SCD1, stearoyl coenzyme A desaturase 1; FAS, fatty acid synthase; PPARα, peroxisome proliferator-activated receptor α; CPT-1, carnitine palmitoyl transferase-1.

**Table 7 plants-11-00607-t007:** Effects of the water extract of fermented rice bran (FRB) on fecal short-chain fatty acid (SCFA) concentrations in aged rats with high-fat (HF) diet feeding.

SCFAs (μM)	NC	HF	HF + 1% FRB	HF + 5% FRB
Propionic acid	55.28 ± 34.73	27.50 ± 24.33	32.33 ± 11.02	43.34 ± 27.99
Isobutyric acid	116.43 ± 4.45	120.84 ± 10.61	122.40 ± 14.67	124.08 ± 8.86
Butyric acid	91.22 ± 23.26	74.47 ± 62.19	52.00 ± 15.29	64.25 ± 22.99
Isovaleric acid	14.30 ± 7.36	13.63 ± 11.76	12.39 ± 2.13	15.16 ± 7.11
Valeric acid	29.44 ± 30.24	13.69 ± 10.47	11.43 ± 3.20	13.60 ± 4.66
4-Methylvaleric acid	1.79 ± 0.20	1.77 ± 0.53	1.81 ± 0.55	2.42 ± 1.27
Hexanoic acid	1.43 ± 0.55	1.16 ± 0.41	0.87 ± 0.42	1.40 ± 1.35
Heptanoic acid	1.23 ± 0.32	0.48 ± 0.26 *	0.50 ± 0.39	0.53 ± 0.15

Values are presented as the mean ± standard deviation (*n* = 4). * *p* < 0.05 vs. the normal control (NC) group; significance between two groups was determined using Student’s *t*-test. In the HF diet-fed groups, significant differences between groups were determined by a one-way ANOVA with Fisher‘s post hoc test.

**Table 8 plants-11-00607-t008:** Composition of the experimental diets.

Ingredient (g/kg)	NC	HF	HF + 1% FRB	HF + 5% FRB
Cornstarch ^1^	465	0	0	0
Maltodextrin ^2^	155	125	122.65	113.258
Sucrose ^3^	100	68.8	67.81	63.86
Casein ^4^	140	200	198.82	194.11
L-cysteine ^5^	2	3	3	3
Fresh soybean oil ^6^	40	25	24.62	23.1
Lard ^7^	0	245	245	245
Cellulose ^8^	50	50	50	50
Mineral mixture (AIN-93M-MIX) ^9^	35	35	33.404	27.02
Vitamin mixture (AIN-93M-MIX) ^10^	10	10	10	10
Choline bitartrate ^11^	3	3	3	3
Tert-butylhydroquinone ^12^	0.008	0.008	0.008	38
Water extract of fermented rice bran (FRB)	0	0	7.6	38
kcal/g	3.808	5.25	5.25	5.22

NC, normal control group; HF, high-fat diet group; HF + 1% FRB, high-fat diet+1% FRB *w/w* group; HF + 5% FRB, high-fat diet+5% FRB *w/w* group. ^1^ Cornstarch: 902956, MP Biomedicals, Irvine, CA, USA; ^2^ maltodextrin: 960429, MP Biochemicals; ^3^ sucrose: Taiwan Sugar Corporation, Taipei, Taiwan; ^4^ casein: 901293, MP Biochemicals; ^5^ l-cysteine: 101454, MP Biochemicals; ^6^ fresh soybean oil: Taiwan Sugar Corporation; ^7^ lard: 902140, MP Biochemicals; ^8^ cellulose: 900453, MP Biochemicals; ^9^ mineral mixture (AIN-93M-MIX): 960401, MP Biochemicals; ^10^ vitamin mixture (AIN-93M-MIX): 2960402, MP Biochemicals; ^11^ choline bitartrate: 101384, MP Biochemicals; ^12^ tert-Butylhydroquinone: 195590, MP Biochemicals.

**Table 9 plants-11-00607-t009:** Antibodies used for Western blotting.

	Antibody (Ab)	Ab Type	Product No.	Source
Primary antibody	SIRT1	monoclonal	#9475	Cell Signaling Technology
	AMPKα	polyclonal	#2532	Cell Signaling Technology
	p-AMPKα	monoclonal	#2535	Cell Signaling Technology
	Leptin receptor	polyclonal	DF7139	Affinity Biosciences
	AdipoR2	polyclonal	DF12811	Affinity Biosciences
Internal control	GAPDH	monoclonal	HRP-60004	Proteintech
Secondary antibody	anti-rabbit IgG		C04003	Croyez Bioscience

SIRT1, NAD-dependent sirtuin-1; AMPKα, adenosine monophosphate-activated protein kinase-α; p-AMPKα, phosphorylated-AMPKα; AdipoR2, adiponectin receptor 2.

**Table 10 plants-11-00607-t010:** Primers used for the quantitative polymerase chain reaction.

	Forward 5′→3′	Reverse 5′→3′
SREBP-1c	AGGAGGCCATCTTGTTGCTT	GTTTTGACCCTTAGGGCAGC
ACC	GGAAGACCTGGTCAAGAAGAAAAT	CACCAGATCCTTATTATTGT
SCD1	GTTGGGTGCCTTATCGCTTTCC	CTCCAGCCAGCCTCTTGTCTAC
FAS	CGGCGTGTGATGGGGCTGGTA	AGGAGTAGTAGGCGGTGGTGTAGA
PPARα	CGGGTCATACTCGCAGGAAA	AAGCGTCTTCTCAGCCATGC
CPT-1	GCATCCCAGGCAAAGAGACA	CGAGCCCTCATAGAGCCAGA
ZO-1	CTTGCCACACTGTGACCCTA	ACAGTTGGCTCCAACAAGGT
Occludin	CTGTCTATGCTCGTCATCG	CATTCCCGATCTAATGACGC
Claudin-1	AAACTCCGCTTTCTGCACCT	TTTGCGAAACGCAGGACATC
β-Actin	CACCAGTTCGCCATGGATGACGA	CCATCACACCCTGGTGCCTAGGGC

SREBP-1c, sterol response element-binding protein-1c; ACC, acetyl CoA carboxylase; SCD1, stearoyl coenzyme A desaturase 1; FAS, fatty acid synthase; PPARα, peroxisome proliferator-activated receptor α; CPT-1, carnitine palmitoyl transferase-1; ZO-1, zonula occludens-1.

## Data Availability

Not applicable.

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
