# Peer review of "Effects of the Water Extract of Fermented Rice Bran on Liver Damage and Intestinal Injury in Aged Rats with High-Fat Diet Feeding"

_plants, 2022, doi:10.3390/plants11050607_

Round 1

Reviewer 1 Report

I recommend the Manuscript for publication. 

I do not have any critical sugestions, but due to the large numer of abbreviations the Manuscript is hard to read. 

Reviewer 2 Report

Manuscript Revision

Title: Effects of the water extract of fermented rice bran on liver damage and intestinal injury in aged rats with high-fat diet feeding.

Personally, I find the text interesting. The authors have carried out interesting work on evaluating the possibility of using waste or underused materials from agriculture in the prevention of serious chronic diseases like metabolic syndrome. Furthermore, this study presents the basis for promising future studies. However, the manuscript presents a series of drawbacks that need to be corrected before the manuscript can be published. In the following lines, I will explain the main mistakes found.

Line 27 – 28. Rewrite.

Line 43. Reference missing.

The introduction is very interesting and complete. However, I miss a figure that explains the information given up to line 85. This would make the information much more visual and easier to follow.

Line 93. “among others” instead of “etc.”.

Line 101 – 106. References missing.

Line 107 – 111. First of all, in a scientific manuscript, the use of first-person should be avoided. Moreover, the use of the connectors is not adequate.

The text includes a large number of abbreviations. It could be interesting to collect all of them on a table.

Figure 10 cannot be read. The letter size is too small. Same mistake in figures 11, 12 and 13. Also, would it be possible to use other colors different to red and green? In this way, readers with color blindness could also interpret the figures.

Line 535. Figure 1 appears long after it is first mentioned. Move it.

Line 538 – 544. This information is from the results section. It is not a discussion.

Line 710. “ad libitum” instead of “ad libitum”. Latin expressions should be in italics.

Line 736. “distilled water” instead of “distilled (d)H2O”.

Line 738. dH2O means distilled water? Moreover, it should be dH2O.

Line 783. “mL” instead of “ml”.

Line 804. “×g”. In the rest of the study, the units are rpm. Unify the format.

Line 807. “assay diluent” instead of “Assay Diluent”.

Line 922. It is infrequent to find figures that are part of the conclusion. I don't think this is the right section to put it. I would move it to the results section.

FINAL REMARKS

In my opinion, the authors have carried out a really interesting study, with promising expectations for future research. The manuscript is clear and well written. However, there are some minor issues that should be improved. Therefore, I am suggesting MINOR REVISIONS. The study should be improved before publication.
